# What's Inside Your Diffusion Model? A Score-Based Riemannian Metric to Explore the Data Manifold

## Abstract

Recent advances in diffusion models have demonstrated their remarkable ability to capture complex image distributions, but the geometric properties of the learned data manifold remain poorly understood. We address this gap by introducing a score-based Riemannian metric that leverages the Stein score function from diffusion models to characterize the intrinsic geometry of the data manifold without requiring explicit parameterization. Specifically, our approach defines a metric tensor in the ambient space that stretches distances in directions perpendicular to the manifold while preserving them along tangential directions, effectively creating a geometry where geodesics naturally follow the manifold's contours. We develop efficient algorithms for computing these geodesics and demonstrate their utility for both interpolation between data points and extrapolation beyond the observed data distribution. Through experiments on synthetic data with known geometry, Rotated MNIST, and complex natural images via Stable Diffusion, we show that our score-based geodesics capture meaningful transformations that respect the underlying data distribution. Our method consistently outperforms baseline approaches on perceptual metrics (LPIPS) and distribution-level metrics (FID, KID), producing smoother, more realistic image transitions. These results reveal the implicit geometric structure learned by diffusion models and provide a principled way to navigate the manifold of natural images through the lens of Riemannian geometry.

## 1 Introduction

The geometry of natural images can be studied by viewing them as points on a manifold in high-dimensional ambient – pixel – space. This perspective, commonly known as the manifold hypothesis (Tenenbaum et al., 2000; Bengio et al., 2013), suggests that despite the vast dimensionality of pixel space, natural images concentrate near a much lower-dimensional structure (Carlsson et al., 2008; Henaff et al., 2014). Understanding this geometric structure is crucial for numerous applications including image synthesis, manipulation, and analysis (Goodfellow et al., 2020; Zhu et al., 2017; Mohan et al., 2019), visual perception (DiCarlo & Cox, 2007; Chung et al., 2018), and representation learning (Kingma et al., 2013; Cohen et al., 2020; Chen et al., 2020).

In this work, we propose a *data-dependent metric* derived from the Stein score function, which can be obtained from a diffusion model trained on a given set of images. Recent advances in diffusion models have demonstrated remarkable capabilities in capturing complex image distributions (Ho et al., 2020; Dhariwal & Nichol, 2021), and several approaches have explored data-dependent metrics to uncover the geometrical properties of image manifolds (Kapusniak et al., 2024; Stanczuk et al., 2024; wan, 2021; Diepeveen et al., 2024; Yun et al.; Park et al., 2023; Samuel et al., 2023). However, these previous methods often require explicit manifold parameterization, are computationally intractable for high-dimensional data, or fail to capture the intrinsic geometric structure that reflects perceptual relationships in the data.

Our approach defines a Riemannian metric in the ambient – pixel – space, leveraging score functions from diffusion models to create a geometry that stretches perpendicular to the data manifold while preserving distances along it. This metric allows us to compute distances and geodesic paths between

images directly in pixel space, capturing meaningful relationships that respect the underlying data distribution. Unlike the Euclidean distance, which treats all directions in pixel space equally, our score-based metric accounts for the anisotropic nature of the image manifold, heavily penalizing movement in directions orthogonal to the manifold while preserving natural movement along tangential directions. By generating images along these geodesics, by exploiting a diffusion model, we can visualize and validate these transformations, providing unique insights into the structure of the learned image manifold.

The main contributions of our work are: **(i)** A novel score-based Riemannian metric that captures the data manifold geometry without explicit parameterization; **(ii)** Efficient algorithms for computing geodesics and manifold extrapolation in high-dimensional image space; **(iii)** Empirical validation on both synthetic data with known geometry and complex image data.

## 2 BACKGROUND

### 2.1 THE MANIFOLD HYPOTHESIS FOR NATURAL IMAGES

The manifold hypothesis posits that high-dimensional data, such as natural images, lie approximately on a low-dimensional manifold embedded in the ambient space $\mathbb{R}^N$ (Tenenbaum et al., 2000; Bengio et al., 2013; Roweis & Saul, 2000). In this framework, images are represented as points in a $N$-dimensional pixel space, where each axis represents a pixel value. Despite the vast dimensionality of this space – often millions of dimensions for modern high-resolution images –, natural images occupy only a tiny fraction of it due to strong correlations and constraints that restrict the set of plausible pixel configurations (Carlsson et al., 2008; Henaff et al., 2014).

Each set of images can be characterized by its own probability density function $p(\boldsymbol{x})$ defined over the pixel space $\mathbb{R}^N$. The key insight is that $p(\boldsymbol{x})$ concentrates its mass on a much lower-dimensional structure than the ambient dimension $N$, referred to as the *support* of the data distribution. Under the manifold hypothesis, this support approximates a lower-dimensional manifold $\mathcal{M} \subset \mathbb{R}^N$ (Figure 1 A). To capture the geometry of this implicit manifold, we employ *data-induced metrics* - Riemannian metrics defined in the ambient space that adapt locally according to the underlying probability density (Lee, 1997). Our approach explores how ambient space deformation enables geodesics that naturally follow the manifold's contours (Figure 1 B,C).

### 2.2 DIFFUSION MODELS & STEIN SCORE

Diffusion models have emerged as powerful generative approaches that gradually transform noise into structured data through an iterative denoising process (Sohl-Dickstein et al., 2015; Ho et al., 2020). These models define a forward process $q$ that progressively adds Gaussian noise to data points $\mathbf{x}_0 \sim p(\mathbf{x})$ according to a predefined schedule, creating a sequence of increasingly noisy versions $\mathbf{x}_t$ with $t$ representing the diffusion step:

$$q(\mathbf{x}_t|\mathbf{x}_0) = \mathcal{N}(\mathbf{x}_t; \sqrt{\bar{\alpha}_t}\mathbf{x}_0, (1 - \bar{\alpha}_t)\mathbf{I}) \tag{1}$$

where $\bar{\alpha}_t$ represents cumulative noise schedule parameters. This process transforms any complex data distribution into a simple Gaussian distribution at the limit $t \to T$. The conditional distribution can be equivalently expressed in a sampling form, which makes the noise component explicit:

$$\mathbf{x}_t = \sqrt{\bar{\alpha}_t}\mathbf{x}_0 + \sqrt{1 - \bar{\alpha}_t}\boldsymbol{\epsilon}, \quad \boldsymbol{\epsilon} \sim \mathcal{N}(\mathbf{0}, \mathbf{I}) \tag{2}$$

where $\boldsymbol{\epsilon}$ is the standard Gaussian noise added during the forward diffusion process. This formulation highlights that at any timestep $t$, the noisy sample $\mathbf{x}_t$ is a combination of the original data point $\mathbf{x}_0$ and noise $\boldsymbol{\epsilon}$, with their proportions determined by the noise schedule.

The score function $\mathbf{s}(\mathbf{x}) = \nabla_{\mathbf{x}} \log p(\mathbf{x})$ is central to diffusion models, which represents the gradient of the log probability density. For data concentrated on a low-dimensional manifold $\mathcal{M} \subset \mathbb{R}^N$, the score function has a crucial geometric interpretation: it points approximately normal to the data manifold, with magnitude increasing with distance from the manifold (Stanczuk et al., 2024; Pidstrigach, 2022; Yun et al.). When training diffusion models, the neural network is typically trained to predict the noise $\boldsymbol{\epsilon}$ that was added during the forward process. This prediction, denoted as $\boldsymbol{\epsilon}_\theta(\mathbf{x}_t, t)$, aims to approximate the true noise $\boldsymbol{\epsilon}$ used to generate $\mathbf{x}_t$ from $\mathbf{x}_0$. Importantly, this noise prediction

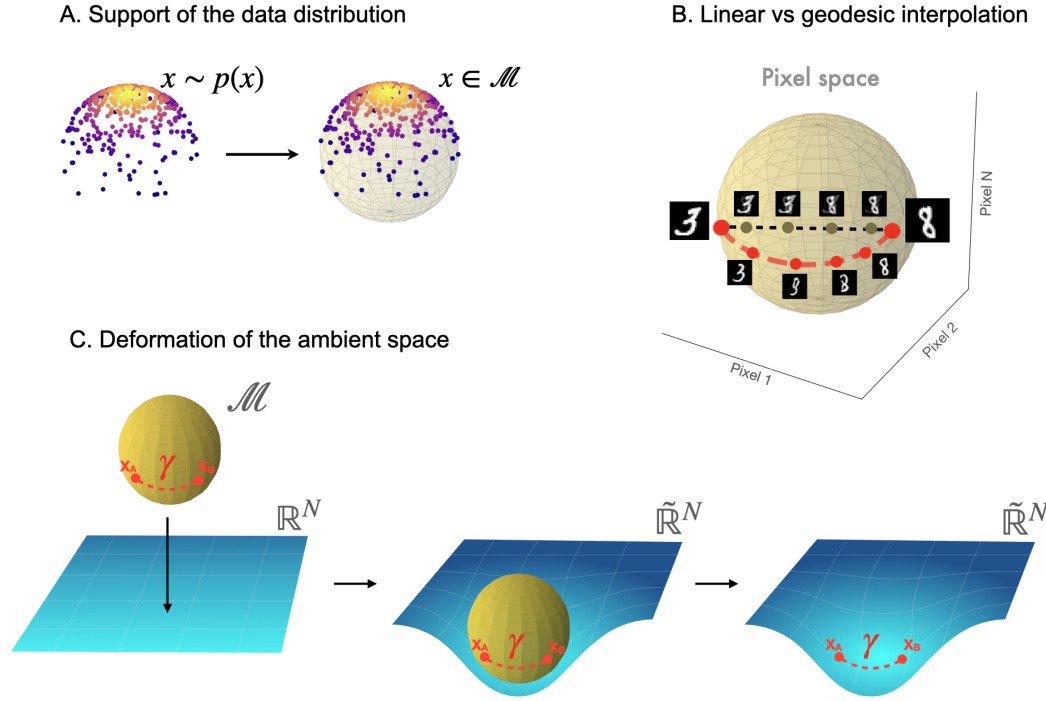

Figure 1: (A) Data points $\mathbf{x}$ sampled from a probability distribution $p(\mathbf{x})$ (left) concentrate on a lower-dimensional manifold $\mathcal{M}$ (right). (B) Linear interpolation (black dashed) versus geodesic interpolation (red curve) between MNIST digits. Geodesics follow the manifold surface, producing valid digit transitions, while linear paths yield superpositions. (C) Geometric deformation (Thorne et al., 2000): data manifold $\mathcal{M}$ embedded in $\mathbb{R}^N$ (left) transforms flat Euclidean space into curved metric space $\tilde{\mathbb{R}}^N$ (middle, right). In this deformed space, geodesics of $\mathcal{M}$ can be computed directly as geodesics of $\tilde{\mathbb{R}}^N$.

directly relates to the score of the noisy distribution through:

$$\mathbf{s}_t(\mathbf{x}_t) = \nabla_{\mathbf{x}_t} \log p_t(\mathbf{x}_t) = -\frac{\boldsymbol{\epsilon}_\theta(\mathbf{x}_t, t)}{\sigma_t} \tag{3}$$

where $\sigma_t = \sqrt{1 - \bar{\alpha}_t}$ is the standard deviation of the noise at timestep $t$. This relationship, derived from the Tweedie-Robbins-Miyasawa formula (Robbins, 1956; Miyasawa et al., 1961), enables us to extract geometric information about the underlying data manifold without requiring explicit parameterization.

## 3 METHODS

### 3.1 CAPTURING MANIFOLD GEOMETRY VIA AMBIENT METRIC DEFORMATION: THE STEIN SCORE METRIC TENSOR

*"Space tells matter how to move; matter tells spacetime how to curve"* C.W. Misner et al. (Thorne et al., 2000)

We define a data-induced metric through geometric deformation (Simon, 2002; Ehrlich, 1974), drawing inspiration from general relativity, where mass curves spacetime to create gravitational paths. Analogously, our approach deforms the ambient Euclidean space such that geodesics naturally adhere to the data manifold without requiring explicit parameterization (Figure 1 C). Rather than directly characterizing the manifold $\mathcal{M}$, we instead deform the geometry of $\mathbb{R}^N$ to accommodate the manifold's presence—effectively creating a "gravitational pull" toward the data distribution.

To formalize this intuition, we leverage score vectors (normal to the manifold) to indirectly characterize the geometry.

**[Definition]** The *Stein score metric tensor* is a smooth map $g : \mathbb{R}^N \to \mathrm{SPD}(N)$ (the space of symmetric positive-definite $N \times N$ matrices) defined at a point $\boldsymbol{x} \in \mathbb{R}^N$ as:

$$g(\boldsymbol{x}) = \mathbf{I} + \lambda \cdot \mathbf{s}(\boldsymbol{x})\mathbf{s}(\boldsymbol{x})^T, \tag{4}$$

where $\mathbf{I}$ is the $N \times N$ identity matrix, $\mathbf{s}(\boldsymbol{x}) = \nabla_{\boldsymbol{x}} \log p(\boldsymbol{x})$ is the score function, and $\lambda$ is a positive penalty parameter that controls the strength of penalization along normal directions (see Appendix D.2 for discussion on optimal $\lambda$).

The Stein score metric $g(\boldsymbol{x})$ equips the ambient space with a Riemannian structure respecting the underlying data manifold geometry. The inner product between vectors $\boldsymbol{u}, \boldsymbol{v} \in \mathbb{R}^N$ at point $\boldsymbol{x}$ becomes: $< \boldsymbol{u}, \boldsymbol{v} >_{g(\boldsymbol{x})} = \boldsymbol{u}^T \boldsymbol{v} + \lambda(\mathbf{s}(\boldsymbol{x})^T \boldsymbol{u})(\mathbf{s}(\boldsymbol{x})^T \boldsymbol{v})$, combining the standard Euclidean inner product with a term that penalizes movement normal to the manifold.

This construction has several desirable properties:

1. When $\lambda = 0$, we recover the standard Euclidean metric;
2. As $\lambda$ increases, the metric becomes increasingly stretched in the direction of the score vector;
3. The metric remains positive definite for all values of $\lambda$ (see Appendix B).

Note that our metric construction finds theoretical grounding in the framework of rank-1 metric perturbations introduced by Hartmann et al. (2022), who demonstrated that metrics of the form $G(x) = I + \lambda \nabla f(x) \nabla f(x)^T$, where $f(x) = \log p(x)$ create anisotropic Riemannian structures that preferentially penalize movement along gradient directions (see Appendix A).

## 3.2 GEOMETRIC COMPUTATION ON THE DATA MANIFOLD

We now develop the mathematical foundation for understanding distances and shortest paths in our deformed ambient space.

**Length of a curve and Energy functionals**. A *curve* in the ambient space $\mathbb{R}^N$ is a smooth function $\gamma : [0, 1] \to \mathbb{R}^N$. The *length* of this curve under our score-based metric is computed as:

$$\mathcal{L}[\gamma] = \int_0^1 \|\dot{\gamma}(\tau)\|_{g(\gamma(\tau))} d\tau = \int_0^1 \sqrt{\dot{\gamma}(\tau)^T \dot{\gamma}(\tau) + \lambda(\mathbf{s}(\gamma(\tau))^T \dot{\gamma}(\tau))^2} \, d\tau, \tag{5}$$

where $\dot{\gamma}(\tau) = \frac{d}{d\tau}\gamma(\tau)$ is the velocity vector of the curve at parameter $\tau$. This length functional measures the total distance traveled along the curve, accounting for the anisotropic nature of our metric.

The *energy* of a curve, which is often more convenient for computational purposes, is defined as:

$$\mathcal{E}[\gamma] = \frac{1}{2}\int_0^1 \|\dot{\gamma}(\tau)\|_{g(\gamma(\tau))}^2 d\tau = \frac{1}{2}\int_0^1 \left[\|\dot{\gamma}(\tau)\|^2 + \lambda(\mathbf{s}(\gamma(\tau))^T \dot{\gamma}(\tau))^2\right] d\tau. \tag{6}$$

This energy functional penalizes curves that move in directions normal to the data manifold (when $\mathbf{s}(\gamma(\tau))^T \dot{\gamma}(\tau) \neq 0$) while imposing no additional cost for movement along tangential directions. We choose to optimize the energy functional rather than length because it eliminates the square root operation, making numerical optimization more stable.

**Geodesics**. *Geodesics* are curves with minimal length connecting two points in a Riemannian manifold. In our score-based metric space, geodesics are fundamental as they provide the optimal paths between points while naturally respecting the underlying data manifold structure.

Formally, a geodesic in the ambient space $\mathbb{R}^N$ is a curve with minimal length between two points $\mathbf{x}_A$ and $\mathbf{x}_B \in \mathbb{R}^N$. To find the shortest path between these points, we solve for the curve that minimizes the length or, equivalently and more conveniently, the energy functional

$$\gamma^* = \arg\min_{\substack{\gamma \\ \gamma(0)=\mathbf{x}_A, \gamma(1)=\mathbf{x}_B}} \mathcal{E}[\gamma] = \arg\min_{\substack{\gamma \\ \gamma(0)=\mathbf{x}_A, \gamma(1)=\mathbf{x}_B}} \frac{1}{2}\int_0^1 \|\dot{\gamma}(\tau)\|_{g(\gamma(\tau))}^2 d\tau, \tag{7}$$

where $\gamma^* := \gamma^*(\tau)$ represents the optimal geodesic curve connecting $\mathbf{x}_A$ and $\mathbf{x}_B$.

**Geodesic Computation Algorithm**. While the geodesic equation provides a mathematical foundation, directly solving this minimization problem in high-dimensional spaces is challenging due to the computational complexity and the irregular behavior of score functions in data-sparse regions. Our approach leverages diffusion models to address these challenges through a three-stage process.

First, we apply controlled noise perturbation by sampling $\boldsymbol{\epsilon} \sim \mathcal{N}(0, \mathbf{I})$ and computing $\mathbf{x}_t = \sqrt{\bar{\alpha}_t}\mathbf{x}_0 + \sqrt{1 - \bar{\alpha}_t}\boldsymbol{\epsilon}$ for both endpoints. This forward diffusion process serves two purposes: it smooths the optimization landscape by regularizing the energy functional and ensures that identical noise is applied to both endpoints, preserving their relative positioning while making the optimization more stable. The noise level $t$ is a hyperparameter that controls the smoothness of the optimization landscape—higher values provide more stability but potentially less accurate manifold adherence.

With the score function $\mathbf{s}(\mathbf{x})$ obtained from the diffusion model at timestep $t$, we construct our metric tensor (Eq. 4) using a fixed scale parameter $\lambda$. The optimal choice of $\lambda$ depends on both the dataset characteristics and the specific noise level $t$ (see Section 4). Next, we discretize the energy functional and solve it numerically (for more details see E). We represent the path as a sequence of points $\gamma = \{\gamma_0, \gamma_1, ..., \gamma_n\}$, with $\gamma_0 = \mathbf{x}_A$ and $\gamma_n = \mathbf{x}_B$. The discrete energy becomes:

$$\mathcal{E}[\gamma] \approx \frac{1}{2}\sum_{i=0}^{n-1} \|(\gamma_{i+1} - \gamma_i)\|^2_{g(\gamma_i)} = \frac{1}{2}\sum_{i=0}^{n-1} \left[\|(\gamma_{i+1} - \gamma_i)\|^2 + \lambda(\mathbf{s}(\gamma_i)^T(\gamma_{i+1} - \gamma_i))^2\right] \quad (8)$$

We minimize this energy using Riemannian gradient descent methods (Bonnabel, 2013) that respect the curved geometry defined by our metric (for more details see Appendix E and Algorithm 2). The gradient computation takes into account how changes in each interior point affect both its contribution to the energy and the score-based metric at that point. We initialize the path with linear interpolation between the endpoints and iteratively refine it until convergence or a maximum iteration count is reached (see Appendix for experiment specific details). Finally, we map the optimized path in noise space back to image space using the denoising capabilities of the diffusion model. Each point along the geodesic is denoised from timestep $t$ back to $t' = 0$.

The three-stage process—noise addition, geodesic optimization, and denoising—creates paths that naturally follow the data manifold while remaining computationally tractable for high-dimensional image data.

**Manifold-Aware Interpolation**. Armed with our score-based metric tensor and geodesic computation algorithm, we can now perform interpolation between data points that respects the underlying manifold structure. Unlike conventional approaches such as linear interpolation (LERP) (Ho et al., 2020), spherical interpolation (SLERP) (Shoemake, 1985; Song et al., 2020a;b) or Noise Diffusion (Zheng et al., 2024) that often cut through low-density regions of the data space, our geodesic-based interpolation follows the natural contours of the data manifold. This quality improvement (see Section 4) stems from the path's adherence to the manifold structure, which prevents interpolation through low-density "off-manifold" regions where the diffusion model has not been trained.

**Manifold-Aware Extrapolation**. While interpolation connects two known endpoints, extrapolation extends a path beyond the observed data, requiring a different approach that respects the manifold without a target endpoint. We implement manifold-aware extrapolation through guided walking by the score-metric tensor with momentum.

Given a geodesic path ending at point $\mathbf{x}_B$, we extrapolate by iteratively computing new points:

$$\mathbf{x}_{i+1} = \mathbf{x}_i + \mathbf{d}_i \quad (9)$$

where $\mathbf{d}_i$ is a direction vector computed as a weighted combination of three components:

$$\mathbf{d}_i = (1 - \varepsilon) \cdot \mathbf{m}_i + \varepsilon \cdot \mathbf{s}(\mathbf{x}_i) \quad (10)$$

Here $\mathbf{m}_i$ is the momentum term that maintains trajectory consistency, $\mathbf{s}(\mathbf{x}_i)$ is the score function guiding the path toward the data manifold. The $\varepsilon$ parameter controls the influence of manifold guidance. The momentum vector is updated using an exponential moving average:

$$\mathbf{m}_{i+1} = \beta \cdot \mathbf{m}_i + (1 - \beta) \cdot \mathbf{d}_i \quad (11)$$

where $\beta$ controls how strongly the extrapolation maintains its previous direction. We initialize $\mathbf{m}_0$ from the tangent direction at the endpoint of the geodesic path (for more details see H). This approach

offers several advantages: it maintains coherent progression by preserving directional momentum; it adheres to the manifold through score guidance. With computational complexity of $\mathcal{O}(n \cdot d)$ per step and requiring only one score evaluation per iteration.

## 4 RESULTS

### 4.1 EMBEDDED SPHERE

To validate our approach, we first consider a 2-sphere embedded in $\mathbb{R}^{100}$. We generate samples from a von Mises-Fisher distribution on $\mathbb{S}^2 \subset \mathbb{R}^3$ (Figure 2A) and construct an isometric embedding – through QR decomposition – into $\mathbb{R}^{100}$, with points representable as 10×10 pixel images (Figure 2B). This controlled testbed offers analytical tractability while mimicking high-dimensional image data.

We train a diffusion model on the embedded samples and extract the score function $\mathbf{s}(\mathbf{x})$. As shown in Figure 2C, the learned score vectors align with normal vectors of the sphere (quantitative validation in C), replicating the theoretical prediction that score functions are normal to the data manifold (Stanczuk et al., 2024). Using these score functions, we compute geodesics and compare with linear interpolation. Figure 2D-F shows that while linear paths cut through the sphere (producing blurred intermediate images), our geodesics follow the manifold's surface (maintaining high sample quality throughout). Similarly, for extrapolation, our approach follows the sphere's curvature and preserves image structure, while linear extrapolation quickly departs from the manifold, causing severe distortions.

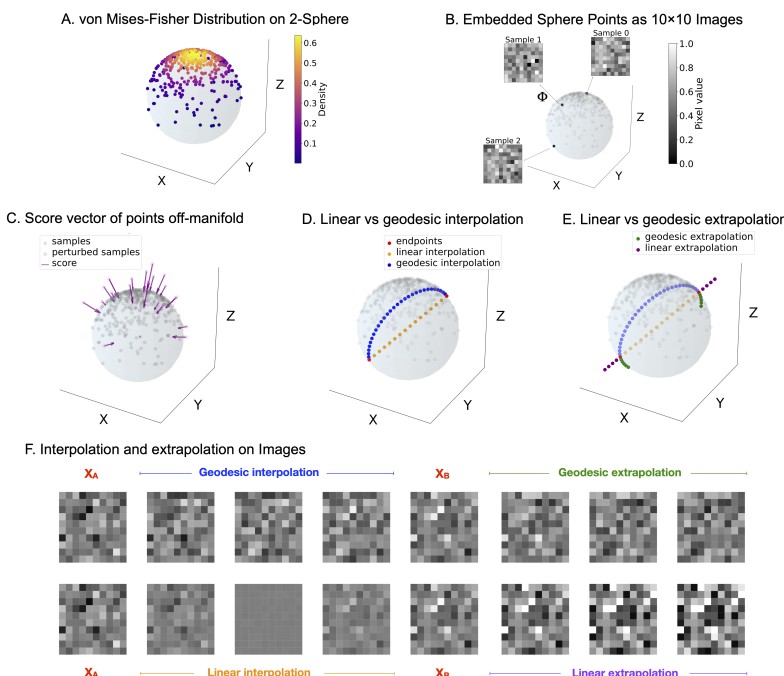

Figure 2: Embedded sphere experiments: (A) von Mises-Fisher distribution on a 2-sphere. (B) Example 10×10 pixel images from the embedding. (C) Score vectors (purple) point normally to the manifold. (D) Linear interpolation (orange) versus geodesics (blue) between endpoints. (E) Extrapolation comparison with geodesics (purple) following the manifold versus linear paths (green) departing from it. (F) Image space results showing our geodesic paths maintain sample quality while linear paths produce blurring (interpolation) or artifacts (extrapolation).

## 4.2 ROTATED MNIST

To evaluate whether our score-based metric captures perceptually meaningful transformations, we conducted experiments on a modified version of the MNIST dataset where the transformation structure is well-defined. We created a custom Rotated MNIST dataset by systematically rotating each digit by angles ranging from $0°$ to $350°$ in $10°$ increments, producing 36 rotated versions of each original digit. We then trained a denoising diffusion model on this dataset using a UNet-2D (Ronneberger et al., 2015) architecture with attention blocks. The model was trained for 100 epochs with a batch size of 256 (see more details in Appendix). Our hypothesis is that this trained model implicitly learns the data manifold, including the rotational structure of the dataset.

For interpolation experiments, we selected test set digit pairs with varying orientations, typically choosing the same digit rendered at different angles. Figure 3A compares our geodesic interpolation method against LERP, SLERP and Noise Diffusion. The results show that our geodesic approach produces trajectories that follow the rotational structure of the data manifold, creating smooth transformations that preserve digit identity while naturally rotating from one orientation to another. While LERP often produces unrealistic blending artifacts where intermediate frames superimpose features from both orientations rather than rotating continuously, our method follows the natural rotation transformation. Quantitatively, we measure the quality of interpolated frames using both PSNR and SSIM (Wang et al., 2004) metrics (Table 1), with our geodesic approach outperforming other methods in intermediate frame quality. The extrapolation experiments, shown in Figure 3B,

Table 1: Quality metrics on 100 Rotated MNIST test samples.

| Metric | LERP | SLERP | NoiseDiff | Geodesic (Ours) |
|---|---|---|---|---|
| PSNR ↑ | $14.08 \pm 0.12$ | $13.64 \pm 0.13$ | $13.67 \pm 0.08$ | $\mathbf{14.98 \pm 0.12}$ |
| SSIM ↑ | $0.578 \pm 0.006$ | $0.572 \pm 0.007$ | $0.568 \pm 0.005$ | $\mathbf{0.650 \pm 0.006}$ |

demonstrate the power of our manifold-guided approach for extending transformations beyond observed data points. Starting with two orientations of the same digit, we compute the geodesic path between them and then continue the transformation using our extrapolation algorithm. The results show that our method successfully continues the rotational trajectory, generating novel orientations that maintain digit identity and structure despite never having been explicitly observed in this sequence during training.

Figure 3: Rotated MNIST. **A** Interpolation Example (Best LERP by PSNR) comparing LERP, SLERP, Noise Diffusion and Geodesic (our method). **B** Three examples with our extrapolation method.

## 4.3 STABLE DIFFUSION & MORPHBENCH

To demonstrate our method's applicability to state-of-the-art diffusion models, we conducted experiments with Stable Diffusion 2.1 (Rombach et al., 2022), a powerful latent diffusion model trained on billions of image-text pairs. This allowed us to scale our approach on a more complex, higher-resolution image generation task than our previous experiments. Stable Diffusion operates in a

compressed latent space rather than pixel space, making it an interesting test case for our score-based metric. The diffusion process occurs in a 4×64×64 latent space (representing 512×512 pixel, RGB images), where the data manifold has complex geometry reflecting natural image statistics. We apply our methodology by computing geodesics in this latent space using score estimates from the model's denoising U-Net, then mapping the results back to image space using the model's VAE decoder.

For evaluation, we utilize MorphBench (Zhang et al., 2024), a comprehensive benchmark for assessing image morphing capabilities. MorphBench consists of 90 diverse image pairs organized into two categories: (1) metamorphosis between different objects (66 pairs) and (2) animation of the same object with different attributes (24 pairs). This diversity allows us to evaluate our method across varying transformation complexities.

Figure 4 (see Appendix K, for more examples) presents qualitative comparisons between our geodesic method and baseline approaches (LERP, SLERP, and Noise Diffusion). Our method generates smoother, more natural-looking transitions that better preserve image coherence during the morphing process. While LERP and SLERP often produce blurry intermediate frames with unrealistic composites of both source and target images, and Noise Diffusion frequently generates images with artifacts, our geodesic interpolation creates a more natural progression by following meaningful paths on the data manifold.

Table 2 provides a comprehensive quantitative evaluation. Our geodesic method achieves the best performance on LPIPS (0.3582 vs. 0.3613 for LERP), a learned perceptual similarity metric that correlates strongly with human judgments of visual quality (Zhang et al., 2018). We also outperform all baselines on distribution-level metrics, with the lowest FID (140.61 vs. 148.19 for LERP) (Heusel et al., 2017) and KID scores (0.0863 vs. 0.0935 for LERP) (Bińkowski et al., 2018). These results indicate that our interpolated images are both perceptually coherent and maintain high sample quality that better matches the "true" data distribution. While LERP achieves slightly better results on pixel-level metrics (PSNR: 21.28 vs. 20.88; SSIM: 0.6274 vs. 0.6180), it's well-established that these metrics often fail to capture perceptual quality (Zhang et al., 2018), especially for high-resolution natural images.

| Metric | LERP | SLERP | NOISEDIFF | GEODESIC (Ours) |
|---|---|---|---|---|
| avg_SSIM ↑ | **0.627 ± 0.012** | 0.575 ± 0.014 | 0.404 ± 0.012 | 0.618 ± 0.012 |
| avg_PSNR ↑ | **21.28 ± 0.18** | 20.36 ± 0.22 | 16.73 ± 0.20 | 20.88 ± 0.17 |
| avg_LPIPS ↓ | 0.361 ± 0.008 | 0.396 ± 0.008 | 0.048 ± 0.005 | **0.358 ± 0.007** |
| FID ↓ | 148.2 ± 5.0 | 171.5 ± 5.3 | 271.8 ± 4.7 | **140.6 ± 5.1** |
| KID ↓ | 0.094 ± 0.005 | 0.110 ± 0.006 | 0.186 ± 0.006 | **0.086 ± 0.005** |

Table 2: Aggregated metrics for MorphBench across 90 image pairs.

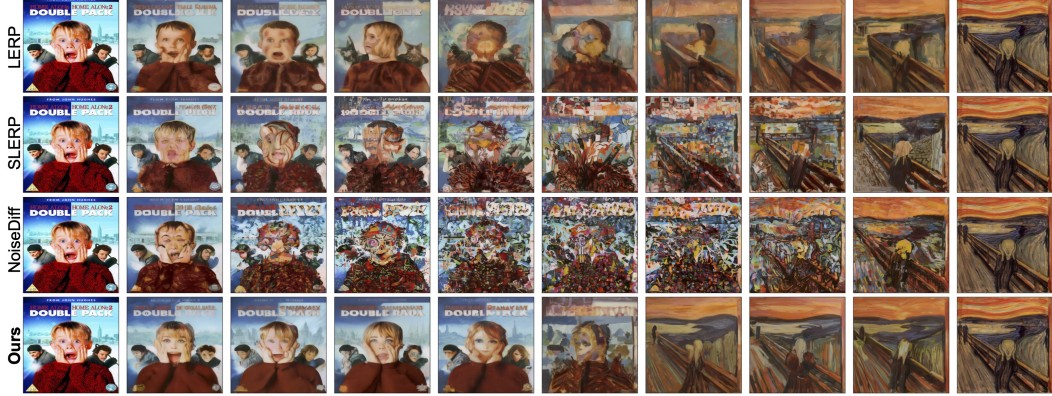

Figure 4: MorphBench interpolation example with Stable Diffusion 2.1. Comparing LERP, SLERP, Noise Diffusion and Geodesic (Our method)

## 5 DISCUSSION

In this work, we introduced a score-based Riemannian metric derived from diffusion models that captures the intrinsic geometry of the data manifold without requiring explicit parameterization. Our approach leverages the Stein score function to define a metric that naturally adapts to the manifold structure, enabling geometric computations that respect the underlying data distribution. Through experiments on synthetic data, MNIST, and complex natural images, we demonstrated that geodesics computed using our score-based metric correspond to perceptually meaningful paths between images, outperforming conventional interpolation methods.

While we used interpolation and extrapolation as validation applications, our approach provides a general framework for understanding and exploring the geometric structure learned by diffusion models. The strong performance on these tasks—despite not being specifically optimized for image morphing—suggests that our score-based metric naturally aligns with human perception of image similarity and transformation paths. This is evidenced by the consistent superiority of our geodesic interpolation approach on perceptual metrics like LPIPS and distribution metrics like FID and KID, indicating that our method produces images that not only look better individually but also follow more natural trajectories through the data space.

An interesting pattern emerged when comparing performance across different datasets. On Rotated MNIST, our method outperformed baselines across pixel-level metrics (PSNR and SSIM). However, on the higher-resolution MorphBench dataset with Stable Diffusion, our method dominated in perceptual metrics (LPIPS) and distribution metrics (FID, KID) but was slightly outperformed by LERP on pixel-level metrics. This difference can be attributed to our geodesic computations for Stable Diffusion occurring in the compressed latent space rather than the pixel space. The VAE decoder introduces additional variability when mapping back to pixels, which may affect pixel-perfect reconstruction while still preserving—and even enhancing—perceptual quality. This observation aligns with the established understanding that pixel-level metrics often fail to capture perceptual similarity in high-resolution natural images, instead favoring blurry but pixel-wise accurate reconstructions over sharper, perceptually preferable images (Zhang et al., 2018).

**Limitations**. Despite its strengths, our approach introduces additional computational complexity compared to direct methods like LERP or SLERP. Our implementation requires solving a non-convex optimization problem with several hundred gradient steps to converge, whereas alternative methods can be computed without any iterative optimization procedure. This reflects an open challenge in Riemannian optimization—efficiently computing geodesics on neural network-defined manifolds (Smith, 2014). As the field of Riemannian optimization advances, we anticipate future improvements that could significantly reduce this computational burden.

A second limitation relates to the validation of extrapolation results. Although our experiments demonstrate that extrapolated images maintain high quality and follow natural extensions of geodesic paths, there are no established quantitative metrics to evaluate extrapolation quality in our settings. This makes objective comparisons challenging, though we view extrapolation primarily as a tool for exploring the implicit structure captured by diffusion models rather than a task requiring ground truth validation.

**Future Work.** One exciting direction would be applying our approach to semantic image editing tasks (Brack et al., 2023; Zhang et al., 2023). By computing geodesic paths between edited and original images, our method could enable more natural and realistic transitions when performing attribute manipulations, style transfers, or object insertions.

Additionally, the computational efficiency of geodesic calculation could be improved through techniques like neural surrogate models (Chen et al., 2018) that directly predict geodesic paths. Such models could enable real-time interactive editing tools that allow artists and designers to explore the learned manifold of a diffusion model while maintaining high sample quality throughout the editing process. Finally, our score-based metric provides a new lens for analyzing the geometry learned by diffusion models.

Future work could explore using this geometric perspective to better understand model behavior, detect biases in learned distributions, or visualize how the data manifold evolves during training. By continuing to develop the connections between diffusion models and Riemannian geometry, we can gain deeper insights into how these powerful generative models represent complex data distributions.

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

## A  RELATED WORKS

**Riemannian Geometry in Diffusion Models.** Recent advances have explored various approaches to construct metrics from generative models and navigating data manifolds. Stanczuk et al. (2024) demonstrated that diffusion models can encode the intrinsic dimension of data manifolds through score function analysis, establishing that score vectors point normal to the data manifold (Theorem 4.1). Building on these insights, Diepeveen et al. (2024) proposed a score-based pullback Riemannian metric that provides closed-form expressions for geodesics and distances on data manifolds. Similarly, Park et al. (2023) analyzed diffusion model latent spaces through pullback metrics derived from the encoding feature maps, enabling semantically meaningful image editing.

For manifold-aware navigation, several approaches have been developed. Kapusniak et al. (2024) proposed Metric Flow Matching (MFM), which learns interpolants by minimizing the kinetic energy of a data-induced Riemannian metric. Samuel et al. (2023) demonstrated that standard interpolation methods like LERP and SLERP produce suboptimal results in high-dimensional latent spaces, proposing norm-guided approaches with non-Euclidean metrics. Complementing interpolation methods, Yun et al. developed techniques for manifold extrapolation using diffusion model denoisers, showing that score functions can guide extrapolation while preserving manifold structure.

**Related Metric Structures.** The rank-1 perturbed metric $g(x) = I + \lambda s(x)s(x)^T$ represents a fundamental construction in differential geometry, with deep connections to deformation theory and general relativity. Llosa & Soler (2005) established that any 3-dimensional Riemannian metric can be locally expressed as $g = a\eta + \epsilon s \otimes s$ (Theorem 1), where $\eta$ is a constant curvature metric and $s$ is a differential 1-form. Coll (1999) generalize this result to arbitrary dimensions, establishing rank-1 perturbations as a canonical form for metric deformation in differential geometry. Remarkably, the Kerr-Schild class of metrics in general relativity meet this relation (Debney et al., 1969).

Beyond theoretical foundations, similar metric constructions have been applied in computational statistics and optimization contexts, and as in Hartmann et al. (2022) (see also (Hartmann et al., 2023; Bergamin et al., 2023; Yu et al., 2023a;b)), are termed the *Monge metrics*. Their construction embeds the target manifold into an extended space using scaled coordinates $\Xi(x) = (x, \alpha \log p(x))$, naturally inducing the metric $G(x) = I + \alpha^2 \nabla \log p(x)\nabla \log p(x)^T$ for efficient MCMC sampling in parameter spaces. This approach creates position-dependent metrics that penalize movement along high-gradient directions, enabling more efficient exploration of complex probability distributions. Our construction follows the same fundamental mathematical principle, using $s(x) = \nabla \log p(x)$ to create directional anisotropy. However, rather than parameter space sampling, we apply this canonical metric form to data manifold exploration, where the score function provides normal directions that enable ambient space geodesic computation without explicit manifold parameterization.

## B  PROOF OF RIEMANNIAN METRIC PROPERTIES

To prove that $g(\boldsymbol{x}) = \mathbf{I} + \lambda \cdot \boldsymbol{s}(\boldsymbol{x})\boldsymbol{s}(\boldsymbol{x})^T$ defines a valid Riemannian metric, we verify the fundamental properties:
**Symmetry:** Immediate since $(\boldsymbol{s}(\boldsymbol{x})\boldsymbol{s}(\boldsymbol{x})^T)^T = \boldsymbol{s}(\boldsymbol{x})\boldsymbol{s}(\boldsymbol{x})^T$;
**Positive-definiteness:** For any non-zero vector $\boldsymbol{v}$:

$$\boldsymbol{v}^T g(\boldsymbol{x})\boldsymbol{v} = \|\boldsymbol{v}\|^2 + \lambda(\boldsymbol{s}(\boldsymbol{x})^T\boldsymbol{v})^2 > 0$$

since $\lambda > 0$ and $\|\boldsymbol{v}\|^2 > 0$ for $\boldsymbol{v} \neq \boldsymbol{0}$.
Thus $g(\boldsymbol{x})$ is a valid Riemannian metric on $\mathbb{R}^N$.

## C  SCORE VECTORS ARE NORMAL TO THE DATA MANIFOLD

To validate the theoretical prediction (Stanczuk et al., 2024) that score vectors align with normal vectors of the data manifold, we implemented a comprehensive statistical analysis framework. For each timestep, we randomly sampled 10,000 points from our spherical dataset and computed the diffusion score function $\mathbf{s}(\mathbf{x})$ using our trained model. Since the data lies on a sphere, the ground truth normal vectors are the radial directions from the origin. We systematically analyzed the geometric relationship between scores and these normals by computing: (1) the angle between each score vector and the corresponding radial direction, (2) variance of the sampled score vectors.

| Timestep | Mean Angle (°) | Angle Variance (°) |
|---|---|---|
| 0 | 127.7 | 1.13 |
| 10 | 129.1 | 1.41 |
| 50 | 167.4 | 1.24 |
| 100 | 170.8 | 1.25 |
| 300 | 172.7 | 1.32 |
| 400 | 172.8 | 1.30 |
| 500 | 172.8 | 1.30 |

Table 3: Quantitative analysis of score-normal alignment across diffusion timesteps. Mean angle approaches 180° as timestep increases, indicating alignment with inward-pointing normals.

As shown in Table 3, the mean angle between score and radial vectors rapidly approaches 173° as the timestep increases, demonstrating that scores strongly align with inward-pointing normals. The consistently small variance values indicate a tightly clustered distribution around the mean angle. This confirms that our approach, which uses $t = 400$ in the main experiments, operates in a regime where score vectors reliably approximate normal vectors to the data manifold, with the consistent inward orientation expected from the maximum likelihood training of diffusion models. These results provide robust empirical validation for the geometric foundation of our metric deformation methodology.

# D    PARAMETERS SELECTION

## D.1    TIME $t$

In *Appendix* C we provide results for understanding when score vectors are normal to the data manifold. We build on this to select $t = 400$, which is further supported by Bodin et al. (2025). The choice of $t \neq T$ has been reported multiple times in the literature (Mokady et al., 2023; Samuel et al., 2023).

## D.2    ABLATION STUDY FOR $\lambda$

Since the true manifold structure for natural images is unknown, we developed a principled approach for selecting the penalty parameter $\lambda$ using controlled experiments on synthetic manifolds with known ground truth geodesics. On a 2-sphere with unitary radius embedded in $\mathbb{R}^{100}$, we studied how $\lambda$ affects geodesic approximation error by comparing our computed geodesics against analytical solutions (see Table 4) .

| $\lambda$ | Relative Error (%) |
|---|---|
| 0 | 4.92 |
| 10 | 0.50 |
| 100 | 0.11 |
| 1000 | 0.07 |
| 5000 | 0.07 |
| 10000 | 0.07 |

Table 4: Performance comparison across different $\lambda$ values

The error decays approximately as $A \cdot \exp(-\lambda/\lambda_0) + B$ (5). Similar results on a 50-sphere in $\mathbb{R}^{100}$ confirmed $\lambda = 1000$ achieves $\sim 0.07\%$ error consistently and after that there are no more improvements. We used this same value for all experiments in the paper.

## D.3    EXTRAPOLATION HYPERPARAMETERS $\beta$ AND $\varepsilon$

For $\beta$ we sticked to the default value (i.e. 0.9) used in Adam (Kingma, 2014) and Riemannian Adam (Becigneul & Ganea, 2019), as the idea of using a moving-average is similar in principle. While for

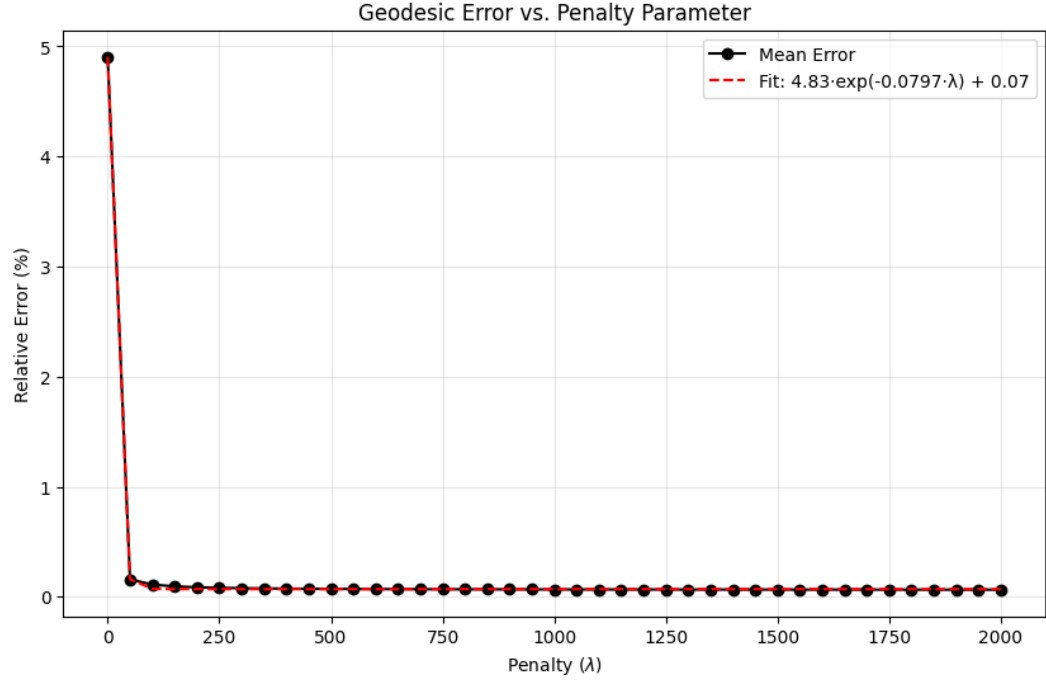

Figure 5: Relative error as a function of penalty parameter $\lambda$. The dashed red line shows the exponential fit $A \exp(-\lambda/\lambda_0) + B$.

$\varepsilon$, we stick to 0.1, giving more importance to the momentum vector. We consider the extrapolation results more as a proof-of-principle and a qualitative results that could enable future works.

## E GEODESIC ALGORITHM DETAILS

### E.1 DISCRETIZATION AND NUMERICAL INTEGRATION

To compute geodesics numerically, we discretize the continuous energy functional using the midpoint rule for integration. This approach provides second-order accuracy and better stability compared to first-order methods such as Euler integration.

Given endpoints $\boldsymbol{x}_A$ and $\boldsymbol{x}_B$, we represent the geodesic path as a sequence of $n + 1$ points:

$$\gamma = \{\gamma_0, \gamma_1, \ldots, \gamma_n\} \tag{12}$$

where $\gamma_0 = \boldsymbol{x}_A$ and $\gamma_n = \boldsymbol{x}_B$ are fixed, and the interior points $\{\gamma_1, \ldots, \gamma_{n-1}\}$ are optimized.

For each segment of the discretized path, we:

1. Compute the velocity vector: $\boldsymbol{v}_i = \gamma_{i+1} - \gamma_i$
2. Calculate the midpoint: $\boldsymbol{m}_i = \frac{1}{2}(\gamma_{i+1} + \gamma_i)$
3. Evaluate the score function at the midpoint: $\boldsymbol{s}_i = s(\boldsymbol{m}_i)$
4. Compute the segment's contribution to the energy functional

The discretized energy functional under the midpoint approximation becomes:

$$\mathcal{E}[\gamma] \approx \frac{1}{2} \sum_{i=0}^{n-1} \|\boldsymbol{v}_i\|^2_{g(\boldsymbol{m}_i)} = \frac{1}{2} \sum_{i=0}^{n-1} \left[ \|\boldsymbol{v}_i\|^2 + \lambda(\boldsymbol{s}_i^T \boldsymbol{v}_i)^2 \right] \tag{13}$$

To improve the optimization stability, we incorporate additional regularization terms:

$$\mathcal{E}_{\text{total}}[\gamma] = \mathcal{E}[\gamma] + \lambda_{\text{smooth}} \mathcal{R}_{\text{smooth}}[\gamma] + \lambda_{\text{mono}} \mathcal{R}_{\text{mono}}[\gamma] \tag{14}$$

where:

$$\mathcal{R}_{\text{smooth}}[\gamma] = \sum_{i=1}^{n-1} \|\gamma_{i+1} - 2\gamma_i + \gamma_{i-1}\|^2 \tag{15}$$

$$\mathcal{R}_{\text{mono}}[\gamma] = \sum_{i=1}^{n-1} \max(0, \|\gamma_i - \gamma_n\| - \|\gamma_{i-1} - \gamma_n\|) \tag{16}$$

The smoothness term penalizes high curvature in the path, while the monotonicity term encourages the path to make consistent progress toward the endpoint.

### E.2 RIEMANNIAN ADAM OPTIMIZATION

To minimize the discretized energy functional, we use a Riemannian extension of the Adam optimizer that properly accounts for the curved geometry defined by our metric. The algorithm proceeds as follows:

---
**Algorithm 1** Riemannian Adam for Geodesic Optimization
---
1: **Input:** Initial path $\gamma = \{\gamma_0, \gamma_1, \dots, \gamma_n\}$ with fixed endpoints
2: **Hyperparameters:** Learning rate $\alpha$, momentum parameters $\beta_1, \beta_2$, stability parameter $\epsilon$
3: **Initialize:** Moment vectors $\boldsymbol{m}_i = \boldsymbol{0}$, $\boldsymbol{v}_i = \boldsymbol{0}$ for $i \in \{1, \dots, n-1\}$
4: $t \leftarrow 0$
5: **while** not converged **do**
6:     $t \leftarrow t + 1$
7:     Compute $\mathcal{E}_{\text{total}}[\gamma]$ and its gradients $\{\nabla_{\gamma_i}\mathcal{E}_{\text{total}}\}_{i=1}^{n-1}$
8:     **for** $i = 1$ to $n - 1$ **do**
9:         $\boldsymbol{g}_i \leftarrow \nabla_{\gamma_i}\mathcal{E}_{\text{total}}$
10:        $\widetilde{\boldsymbol{g}}_i \leftarrow \texttt{RiemannianGradient}(\boldsymbol{g}_i, \gamma_i, g)$ ▷ Convert Euclidean to Riemannian gradient
11:        $\boldsymbol{m}_i \leftarrow \beta_1\boldsymbol{m}_i + (1 - \beta_1)\widetilde{\boldsymbol{g}}_i$
12:        $\boldsymbol{v}_i \leftarrow \beta_2\boldsymbol{v}_i + (1 - \beta_2)\widetilde{\boldsymbol{g}}_i^2$
13:        $\hat{\boldsymbol{m}}_i \leftarrow \boldsymbol{m}_i/(1 - \beta_1^t)$           ▷ Bias correction
14:        $\hat{\boldsymbol{v}}_i \leftarrow \boldsymbol{v}_i/(1 - \beta_2^t)$
15:        $\boldsymbol{\eta}_i \leftarrow \alpha \cdot \hat{\boldsymbol{m}}_i/(\sqrt{\hat{\boldsymbol{v}}_i} + \epsilon)$      ▷ Update direction
16:        $\gamma_i^{\text{old}} \leftarrow \gamma_i$
17:        $\gamma_i \leftarrow \gamma_i - \boldsymbol{\eta}_i$           ▷ Update position
18:        $\boldsymbol{m}_i \leftarrow \texttt{ParallelTransport}(\boldsymbol{m}_i, \gamma_i^{\text{old}}, \gamma_i, g)$   ▷ Transport momentum vector
19:     **end for**
20:     **if** convergence criteria met **then**
21:         **break**
22:     **end if**
23: **end while**
24: **Return** optimized path $\gamma$

---

The key components that extend Adam to Riemannian manifolds are:

**Riemannian Gradient:** The Euclidean gradient $\boldsymbol{g}$ is converted to a Riemannian gradient $\widetilde{\boldsymbol{g}}$ using:

$$\widetilde{\boldsymbol{g}} = g(\boldsymbol{x})^{-1}\boldsymbol{g} \tag{17}$$

For our metric tensor $g(\boldsymbol{x}) = \mathbf{I} + \lambda \cdot \boldsymbol{s}(\boldsymbol{x})\boldsymbol{s}(\boldsymbol{x})^T$, we can efficiently compute this using the Sherman-Morrison formula:

$$\widetilde{\boldsymbol{g}} = \boldsymbol{g} - \frac{\lambda \cdot (\boldsymbol{s}^T\boldsymbol{g}) \cdot \boldsymbol{s}}{1 + \lambda \cdot (\boldsymbol{s}^T\boldsymbol{s})} \tag{18}$$

**Parallel Transport:** To properly preserve momentum information when moving between points on the manifold, we parallel transport the momentum vectors along the update direction:

$$\texttt{ParallelTransport}(\boldsymbol{m}, \boldsymbol{x}_{\text{old}}, \boldsymbol{x}_{\text{new}}, g) = \boldsymbol{m} - \frac{1}{2} \cdot \langle \boldsymbol{m}, \boldsymbol{x}_{\text{new}} - \boldsymbol{x}_{\text{old}} \rangle_g \cdot (\boldsymbol{x}_{\text{new}} - \boldsymbol{x}_{\text{old}}) \tag{19}$$

This first-order approximation of parallel transport is sufficient for our purposes and maintains the directional information of the momentum vectors as they move along the curved manifold.

Through this optimization process, we obtain a discretized geodesic path that minimizes the energy functional while respecting the Riemannian geometry induced by our score-based metric tensor.

## F  GEODESIC COMPUTATION ALGORITHM

We complement the geodesic algorithm description in the main paper with Algorithm 2

---

**Algorithm 2** Geodesic Computation with Score-Based Riemannian Metric

---

1: **Input:** Points $p$, $q$, diffusion timestep $t$, number of segments $n$, scale parameter $\lambda$
2: **Output:** Geodesic path $\gamma = \{\gamma_0, \gamma_1, \ldots, \gamma_n\}$ with $\gamma_0 = p$, $\gamma_n = q$
3: **Forward diffusion stage:**
4: Sample noise $\epsilon \sim \mathcal{N}(0, \mathbf{I})$
5: $\tilde{p} \leftarrow \sqrt{\alpha_t}p + \sqrt{1 - \alpha_t}\epsilon$
6: $\tilde{q} \leftarrow \sqrt{\alpha_t}q + \sqrt{1 - \alpha_t}\epsilon$
7: **Initialize path:**
8: $\gamma_i \leftarrow (1 - \lambda_i)\tilde{p} + \lambda_i\tilde{q}$ for $i \in \{0, 1, \ldots, n\}$, $\lambda_i = i/n$
9: Interior points $\Gamma \leftarrow \{\gamma_1, \gamma_2, \ldots, \gamma_{n-1}\}$
10: **Define score-based metric tensor:**
11: $g_{\boldsymbol{x}}(\boldsymbol{u}, \boldsymbol{v}) = \boldsymbol{u}^T\boldsymbol{v} + \lambda \cdot (\boldsymbol{s}(\boldsymbol{x})^T\boldsymbol{u})(\boldsymbol{s}(\boldsymbol{x})^T\boldsymbol{v})$
12: **Optimize path:**
13: Initialize RiemannianAdam optimizer with interior points $\Gamma$
14: **for** iteration = 1 to max_iterations **do**
15:     // Compute energy using midpoint discretization
16:     $\mathcal{E} \leftarrow 0$
17:     **for** $i = 0$ to $n - 1$ **do**
18:         $\boldsymbol{v}_i \leftarrow \gamma_{i+1} - \gamma_i$         ▷ Segment velocity
19:         $\boldsymbol{m}_i \leftarrow \frac{1}{2}(\gamma_{i+1} + \gamma_i)$         ▷ Segment midpoint
20:         $\mathcal{E} \leftarrow \mathcal{E} + \frac{1}{2}\|\boldsymbol{v}_i\|^2 + \frac{\lambda}{2}(\boldsymbol{s}(\boldsymbol{m}_i)^T\boldsymbol{v}_i)^2$     ▷ Energy contribution
21:     **end for**
22:     // Add regularization terms
23:     $\mathcal{E}_{\text{smooth}} \leftarrow \lambda_{\text{smooth}} \sum_{i=1}^{n-1} \|\gamma_{i+1} - 2\gamma_i + \gamma_{i-1}\|^2$     ▷ Smoothness
24:     $\mathcal{E}_{\text{mono}} \leftarrow \lambda_{\text{mono}} \sum_{i=1}^{n-1} \text{ReLU}(\|\gamma_i - \gamma_n\| - \|\gamma_{i-1} - \gamma_n\|)$     ▷ Monotonicity
25:     $\mathcal{E}_{\text{total}} \leftarrow \mathcal{E} + \mathcal{E}_{\text{smooth}} + \mathcal{E}_{\text{mono}}$
26:     Compute gradients of $\mathcal{E}_{\text{total}}$ with respect to interior points $\Gamma$
27:     Update interior points using Riemannian gradient step
28:     **if** convergence criteria met **then**
29:         **break**
30:     **end if**
31: **end for**
32: **Reverse diffusion:**
33: **for** $i = 0$ to $n$ **do**
34:     **if** $i = 0$ or $i = n$ **then**
35:         $\hat{\gamma}_i \leftarrow$ original clean point ($p$ or $q$)
36:     **else**
37:         $\hat{\gamma}_i \leftarrow \text{Denoise}(\gamma_i, \text{from } t \text{ to } 0)$
38:     **end if**
39: **end for**
40: **return** $\hat{\gamma} = \{\hat{\gamma}_0, \hat{\gamma}_1, \ldots, \hat{\gamma}_n\}$

---

## G  INTERPOLATION

We complement the interpolation algorithm description in the main paper with Algorithm 3

---

**Algorithm 3** Manifold-Aware Interpolation

---
1: **Input:** Points $\boldsymbol{p}$, $\boldsymbol{q}$, number of interpolation points $n$, timestep $t$
2: **Output:** Interpolated sequence $\{\boldsymbol{y}_0, \boldsymbol{y}_1, \ldots, \boldsymbol{y}_n\}$
3: **Stage 1: Forward Diffusion**
4: Sample noise $\boldsymbol{\epsilon} \sim \mathcal{N}(0, \mathbf{I})$               ▷ Same noise for consistency
5: $\tilde{\boldsymbol{p}} \leftarrow \sqrt{\alpha_t}\boldsymbol{p} + \sqrt{1 - \alpha_t}\boldsymbol{\epsilon}$
6: $\tilde{\boldsymbol{q}} \leftarrow \sqrt{\alpha_t}\boldsymbol{q} + \sqrt{1 - \alpha_t}\boldsymbol{\epsilon}$
7: **Stage 2: Geodesic Computation in Noise Space**
8: Set up score extractor $s_\theta$ at timestep $t$
9: Set up Stein metric tensor with adaptive scaling
10: Compute reference scores at endpoints
11: Initialize path via linear interpolation
12: Optimize path using Algorithm 2 (Geodesic Computation)
13: **Stage 3: Backward Diffusion (Denoising)**
14: **for** $i = 0$ to $n$ **do**
15:     **if** $i = 0$ **then**
16:         $\boldsymbol{y}_i \leftarrow \boldsymbol{p}$               ▷ Use original clean endpoint
17:     **else if** $i = n$ **then**
18:         $\boldsymbol{y}_i \leftarrow \boldsymbol{q}$               ▷ Use original clean endpoint
19:     **else**
20:         $\boldsymbol{y}_i \leftarrow \text{DenoisingDiffusion}(\tilde{\gamma}_i, \text{from } t \text{ to } 0)$
21:     **end if**
22: **end for**
23: **return** $\{\boldsymbol{y}_0, \boldsymbol{y}_1, \ldots, \boldsymbol{y}_n\}$

---

## H EXTRAPOLATION

We complement the interpolation algorithm description in the main paper with Algorithm 4

---

**Algorithm 4** Single-Direction Extrapolation

---
1: Initialize $\boldsymbol{x}_{\text{current}} \leftarrow \boldsymbol{q}$
2: Compute initial direction from path segments:
3: $\boldsymbol{m} \leftarrow \sum_{i=1}^{\min(3, n/4)} w_i \cdot (\gamma_{n-i+1} - \gamma_{n-i}) / \sum_i w_i$
4: $\boldsymbol{m} \leftarrow \boldsymbol{m}/\|\boldsymbol{m}\| \cdot \text{step\_size}$
5: **for** $i = 1$ to num\_steps **do**
6:     Compute score at current point: $\boldsymbol{s} \leftarrow \mathbf{s}(\boldsymbol{x}_{\text{current}})$
7:     Compute direction: $\boldsymbol{d} \leftarrow (1 - \varepsilon) \cdot \boldsymbol{m} + \varepsilon \cdot \boldsymbol{s}$
8:     Normalize: $\boldsymbol{d} \leftarrow \boldsymbol{d}/\|\boldsymbol{d}\| \cdot \text{step\_size}$
9:     Update position: $\boldsymbol{x}_{\text{next}} \leftarrow \boldsymbol{x}_{\text{current}} + \boldsymbol{d}$
10:     Update momentum: $\boldsymbol{m} \leftarrow \beta \cdot \boldsymbol{m} + (1 - \beta) \cdot \boldsymbol{d}$
11:     $\boldsymbol{x}_{\text{current}} \leftarrow \boldsymbol{x}_{\text{next}}$
12:     Add $\boldsymbol{x}_{\text{current}}$ to extrapolation path
13: **end for**

---

## I COMPUTATIONAL COMPLEXITY BENCHMARK

We acknowledge that our method is more computationally intensive than baselines (LERP, SLERP, NoiseDiffusion), which are essentially fixed (non-adaptive) computations. However, we believe the computational overhead is reasonable considering the quality improvements and more importantly the data-driven nature.

**Scalability.** Despite the added complexity, our method scales well to large images and datasets. As demonstrated with Stable Diffusion 2.1 on 512×512 images (4×64×64 latents), the metric calculation scales nicely with image (latents) dimensionality.

Here are results averaged over 100 instantiations:

| Method | 5 points | 10 points | 20 points | 30 points | 50 points |
|---|---|---|---|---|---|
| LERP | $0.22 \pm 0.03$ | $0.21 \pm 0.05$ | $0.67 \pm 0.42$ | $0.67 \pm 0.44$ | $0.74 \pm 0.35$ |
| NoiseDiffusion | $0.99 \pm 0.25$ | $1.23 \pm 0.33$ | $2.64 \pm 0.60$ | $3.37 \pm 0.73$ | $4.77 \pm 0.72$ |
| SLERP | $0.99 \pm 0.25$ | $0.89 \pm 0.24$ | $1.04 \pm 0.48$ | $1.11 \pm 0.41$ | $1.08 \pm 0.50$ |
| Ours | $6.16 \pm 0.58$ | $20.7 \pm 9.8$ | $58.1 \pm 10.4$ | $82.8 \pm 10.4$ | $144 \pm 13$ |

Table 5: Computation time for Rotated MNIST (32×32 pixels) in $\times 10^{-3}$ seconds.

| Method | 5 points | 10 points | 20 points | 30 points | 50 points |
|---|---|---|---|---|---|
| LERP | $0.63 \pm 0.33$ | $0.65 \pm 0.31$ | $0.57 \pm 0.32$ | $0.67 \pm 0.38$ | $0.72 \pm 0.42$ |
| NoiseDiffusion | $1.21 \pm 0.62$ | $1.98 \pm 0.73$ | $2.62 \pm 0.63$ | $3.90 \pm 0.81$ | $6.85 \pm 13.52$ |
| SLERP | $1.09 \pm 0.52$ | $1.67 \pm 0.90$ | $1.05 \pm 0.60$ | $0.98 \pm 0.40$ | $1.03 \pm 0.51$ |
| Ours | $14.5 \pm 3.3$ | $30.6 \pm 6.6$ | $58.7 \pm 7.5$ | $85.3 \pm 10.6$ | $146 \pm 14$ |

Table 6: Computation time for Stable Diffusion (4×64×64 latent) in $\times 10^{-3}$ seconds.

**Optimization cost**. The most intensive part is geodesic optimization, which scales linearly with the number of points sampled ($n$) due to the energy functional (Riemannian integral).

**Context**. While baselines are faster, they offer no guarantees of following the learned data manifold. Computing geodesics on implicitly defined manifolds inherently requires optimization—there's no analytical shortcut for complex learned manifolds.

## J    ROTATED MNIST

After setting up the dataset as in Section 4.2, we trained a DDPM (Ho et al., 2020) Model in PyTorch (version 2.7 and cuda 12.6) by exploiting the Diffusers library (von Platen et al., 2022) and Accellerate library (Gugger et al.) to parallelize training on 7 NVIDIA A100 GPUs.

**U-Net Network Architecture**

We employed a U-Net model with the following configuration:

- **Input/Output:** The model accepts 32×32 grayscale images (single channel) and predicts noise with matching dimensions.
- **Depth:** Four resolution levels with downsampling and upsampling operations.
- **Channel Dimensions:** Channel counts of (64, 128, 256, 256) at each respective resolution level.
- **Block Structure:** Each resolution level contains 2 ResNet blocks.
- **Attention Mechanism:** Self-attention blocks at the third level of both encoder and decoder paths to capture global relationships, which is crucial for understanding rotational structure.
- **Downsampling Path:** (DownBlock2D, DownBlock2D, AttnDownBlock2D, Down-Block2D)
- **Upsampling Path:** (UpBlock2D, AttnUpBlock2D, UpBlock2D, UpBlock2D)

The total parameter count of the model is approximately 30 million parameters.

The training procedure was as follows:

- Optimizer: AdamW with learning rate $1 \times 10^{-4}$, $\beta_1 = 0.9$, $\beta_2 = 0.999$
- Learning rate scheduler: Cosine schedule with warmup (500 steps)
- Batch size: 256
- Training duration: 100 epochs
- Loss function: Mean squared error (MSE) between predicted and true noise

- Precision: Mixed precision training with bfloat16
- Training Time: 12 hours

Inference and geodesic optimization was then done on single GPU. For Table 1 we randomly sampled 100 digits from the MNIST test set (not used for training the diffusion models), fixed $t = 400$ diffusion process steps, and run LERP, SLERP, NoiseDiffusion and our method (Geodesic); after that we denoised back to image space and computed PSNR and SSIM.

The geodesic optimization - with 8 points, as in Figure 3 A - took 2 minutes on average per iteration (2000 maximum amount of optimization epochs with 250 patience).

After that we selected a handful of test samples, computed the geodesic interpolation between a reference base angle and this angle $+ 30°$. We then extrapolated for N = 5 steps (results are shown in Figure 3 B). Image extrapolation was extremely fast ($\simeq$ 1 second for 5 steps).

## K  STABLE DIFFUSION & MORPHBENCH

We employed the pre-trained Stable Diffusion 2.1. implementation available in the Diffusers library (von Platen et al., 2022), running with PyTorch 2.7 and cuda 12.6 on a NVIDIA A100 GPU (40GB VRAM).

Geodesic computation with 10 points, 2000 iterations (patience 250 epochs), tipically required 20 mins per image interpolation. We employed t = 400 (diffusion steps)

For our experiments with Stable Diffusion 2.1, we used unconditional generation by providing an empty text prompt (""), allowing the model to focus solely on the image manifold structure without text-based guidance.

**MorphBench Performances**

Examining the method performance across dataset categories reveals interesting patterns. For the animation subset (Table 7 in Appendix), our method excels in perceptual metrics (best LPIPS at 0.3402 and KID at 0.0901), while LERP performs marginally better in pixel metrics and FID. For the more challenging metamorphosis transformations (Table 8 in Appendix), our geodesic approach demonstrates clear advantages in distribution-level metrics (best FID at 145.94 and KID at 0.0848). This superiority in metamorphosis scenarios highlights our method's effectiveness for complex transformations between different objects, where properly navigating the intrinsic manifold geometry is most critical.

Additionally we report more examples of interpolation comparison, see Figure 6, 7, 8

| Metric | LERP | SLERP | NOISEDIFFUSION | GEODESIC |
|---|---|---|---|---|
| avg_ssim ↑ | $0.6251 \pm 0.0108$ | $0.6000 \pm 0.0120$ | $0.4202 \pm 0.0115$ | $\mathbf{0.6261 \pm 0.0103}$ |
| avg_psnr ↑ | $\mathbf{21.23 \pm 0.21}$ | $20.72 \pm 0.24$ | $17.09 \pm 0.23$ | $20.98 \pm 0.21$ |
| avg_lpips ↓ | $0.3467 \pm 0.0075$ | $0.3652 \pm 0.0082$ | $0.4577 \pm 0.0058$ | $\mathbf{0.3402 \pm 0.0068}$ |
| fid ↓ | $\mathbf{125.89 \pm 4.4}$ | $142.08 \pm 4.9$ | $253.52 \pm 5.4$ | $127.29 \pm 5.0$ |
| kid ↓ | $0.0947 \pm 0.0065$ | $0.1059 \pm 0.0072$ | $0.1925 \pm 0.0079$ | $\mathbf{0.0901 \pm 0.0066}$ |

Table 7: Aggregated metrics for Animation dataset across 24 image pairs.

| Metric | LERP | SLERP | NOISEDIFFUSION | GEODESIC |
|---|---|---|---|---|
| avg_ssim ↑ | $\mathbf{0.6282 \pm 0.0129}$ | $0.5653 \pm 0.0152$ | $0.3969 \pm 0.0120$ | $0.6148 \pm 0.0122$ |
| avg_psnr ↑ | $\mathbf{21.29 \pm 0.17}$ | $20.21 \pm 0.22$ | $16.58 \pm 0.19$ | $20.83 \pm 0.17$ |
| avg_lpips ↓ | $0.3672 \pm 0.0077$ | $0.4076 \pm 0.0078$ | $0.4891 \pm 0.0036$ | $\mathbf{0.3653 \pm 0.0069}$ |
| fid ↓ | $157.11 \pm 5.0$ | $183.24 \pm 5.1$ | $279.13 \pm 4.3$ | $\mathbf{145.94 \pm 5.1}$ |
| kid ↓ | $0.0930 \pm 0.0046$ | $0.1120 \pm 0.0049$ | $0.1836 \pm 0.0056$ | $\mathbf{0.0848 \pm 0.0050}$ |

Table 8: Aggregated metrics for Metamorphosis dataset across 66 image pairs.

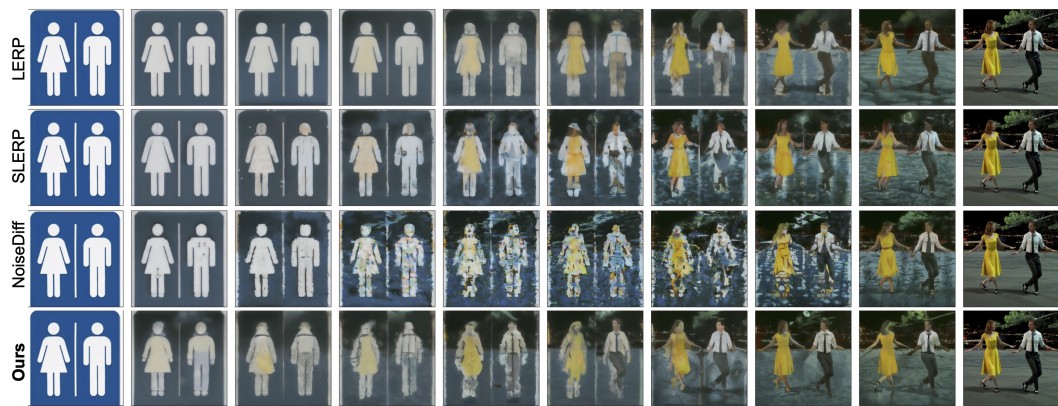

Figure 6: Stable Diffusion. Interpolation Example vs LERP, SLERP and Noise Diffusion.

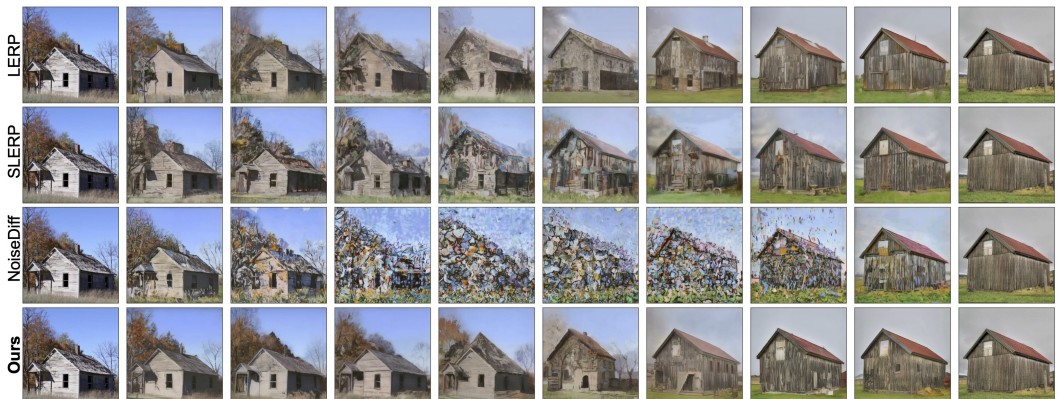

Figure 7: Stable Diffusion. Interpolation Example vs LERP, SLERP and Noise Diffusion.

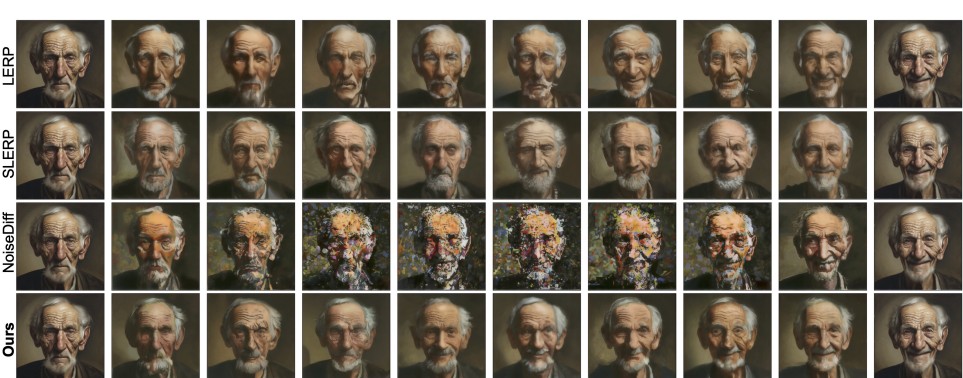

Figure 8: Stable Diffusion. Interpolation Example vs LERP, SLERP and Noise Diffusion.

