# OpenReview forum: "What's Inside Your Diffusion Model? A Score-Based Riemannian Metric to Explore the Data Manifold"
_ICLR.cc/2026/Conference — ICLR 2026 Conference Withdrawn Submission_

### Official Review · Reviewer_7fUT · 2025-10-27

**Soundness:** 1
**Presentation:** 3
**Contribution:** 1
**Rating:** 2
**Confidence:** 4

**Summary:**

This paper proposes a data-driven Riemannian metric built from the score (of a noise-corrupted version) of the data distribution. The proposed Riemannian metric in local coordinates is $G(x)=I+\lambda s(x)s(x)^\top $ where $\lambda>0$ is a regularization parameter. Using this metric, the authors extract data-manifold geodesics by proposing a regularized geodesic optimization algorithm.

**Strengths:**

The paper is well written and provides interesting intuition about constructing a data-driven Riemannian metric that can explain the geometry of the data manifold. They show promising results in uncovering geodesics on simple low-dimensional hyperspheres isometrically embedded in high dimensions, and some improvement relative to linear interpolation on rotated MNIST.

**Weaknesses:**

1. **Theoretical justification.** In Section 3.2 the authors claim, “This energy functional penalizes curves that move in directions normal to the data manifold.” Prior work [3] shows that for small diffusion times the score lies in the normal space, but that does not imply the specific metric $G(x)=I+\lambda s(x)s(x)^\top$ will drive velocities to the tangent bundle. The curve velocity $\dot\gamma$ can still have substantial projection onto the normal space in directions orthogonal to $s(x)$. That orthogonal subspace is $(N-1)$-dimensional, where $N$ is the dimension of the normal space. For instance, let's consider an 1D curve embedded in 3D space. The normal space is 2-dimensional and the tangent space is 1-dimensional. The proposed metric doesn't theoretically penalize movement in the 2D plane orthogonal to $s(x)$. Hence, the proposed rank-1 metric appears theoretically sufficient only when the codimension is $1$. Ideally, the data-driven metric should penalize *all* motion into the normal space.

2. **Lack of sufficient comparison.** The score pullback metric $J^\top J$ penalizes all movement into the normal space. It has been proposed by Diepeveen et al. [1] (for normalizing flows) and by Saito & Matsubara [2] (for diffusion models). In local coordinates, it is written as $G(x)=\lambda I + J(x)^\top J(x)$, where $J$ is the Jacobian of the score $s(x)$ and $\lambda>0$ is a small regularization constant for well-posedness. Earlier work ([3], [4]) indicates that for small diffusion times $J$ has large singular values along normal directions and vanishing ones along tangent directions, so $J^\top J$ makes all movement into the normal space prohibitively costly. Although this metric may seem expensive to compute due to the use of Jacobian, geodesic optimization only needs Jacobian–vector products (efficient in modern DL frameworks), and Diepeveen et al. further leverage pullback-geometry machinery that yields closed-form geodesics. Given their stronger theoretical footing and practical efficiency, these approaches should be included in the benchmarks.

3. **Experiments.** On real images (MorphBench), the method shows no clear advantage over latent-space linear interpolation (LERP), both qualitatively and quantitatively (Table 2, Figure 4). Additional synthetic-manifold studies would strengthen the case—e.g., manifolds with spatially varying curvature in low dimensions (a closed wiggly curve/surface in 2D/3D) to probe where the metric helps or fails.

4. **No exploration of other manifold properties.** It remains unclear whether the proposed metric can reveal further properties such as curvature or support a useful Riemannian projection. For instance, one could test whether $G^{-1}$ acts as a projection operator enabling Riemannian optimization on the data manifold: define a convex objective on a simple manifold with codimension larger than one and verify that optimization with the proposed metric reaches the global minimum while staying on-manifold.

**References**

[1] Willem Diepeveen, Georgios Batzolis, Zakhar Shumaylov, and Carola-Bibiane Schönlieb. *Score-based pullback Riemannian geometry: Extracting the data manifold geometry using anisotropic flows.*

[2] Shinnosuke Saito and Takashi Matsubara. *Image Interpolation with Score-Based Riemannian Metrics of Diffusion Models.*

[3] Jan Paweł Stańczuk, Georgios Batzolis, Teo Deveney, and Carola-Bibiane Schönlieb. *Diffusion Models Encode the Intrinsic Dimension of Data Manifolds.*

[4] Li Kevin Wenliang and Ben Moran. *Score-based generative models learn manifold-like structures with constrained mixing.*

**Questions:**

Q1.) How does the proposed metric guarantee no motion into the normal space in directions orthogonal to s(x)?

Q2.) Shouldn't the choice of the diffusion time t at which the optimization takes place depend on the data distribution? Important geometric information could be destroyed at diffusion time $t=400$ for many real distributions. How should practitioners choose the value t for distributions that are considerably different from the ones considered in this paper?

Regarding Appendix C: on the sphere you perturb points and measure the angle between the score $s(x)$ and the outward radial vector. For small perturbations, points fall both inside and outside the surface, so the angle is bimodal near $0^\circ$ and $180^\circ$; the mean should therefore be $\approx 90^\circ$ (your $\sim 120^\circ$ at small diffusion times likely reflects estimation noise). By contrast, the near-$180^\circ$ mean at $t=400$ suggests you are already in the Gaussian regime where
$p_t \approx \mathcal{N}(0,\sigma_t^2 I)$ and $\nabla \log p_t(x) \approx -\frac{x}{\sigma_t^2},$
i.e., scores are inward radial regardless of the underlying manifold. This indicates that the manifold structure has largely been destroyed by that time which explains why the mean angle is close to 180 degrees from t=300 onwards. For t=1000, you should also see mean angles close to 180 degrees. This analysis raises serious concerns about all your experimental results as it seems that you have chosen diffusion time $t=400$ based on a wrong interpretation and used it consistently for all your experiments. Please correct me if I am making a mistake.

Q3.) Can the proposed metric extract other manifold properties such as curvature or the Riemannian projection operator?

---

### Official Review · Reviewer_hYrq · 2025-10-28

**Soundness:** 3
**Presentation:** 1
**Contribution:** 2
**Rating:** 2
**Confidence:** 4

**Summary:**

This work defines a Riemannian metric from the score function $s$ of a pre-trained diffusion model in order to extract the data manifold. The metric has the form $g = I + \lambda s s^T$. Because the score function is considered to be directed to the data manifold, this acts as a penalty in that direction and suppresses movement away from the manifold. Experiments show that this metric enables natural image interpolation.

**Strengths:**

The proposed approach is simple, intuitive, and theoretically interpretable, compared to existing studies that try to understand the internals of pretrained generative models.

This paper clearly explains the implementation, which makes it run off-the-shelf.

This paper presents both analytical results on synthetic data and practical results on real data.

**Weaknesses:**

### Soundness

(1) This metric uses an extremely large coefficient $\lambda = 1000$, which makes $g$ nearly rank-1. This raises doubts about the validity as a Riemannian metric. When $\lambda$ is small, the method reduces to LERP and is expected to perform poorly. Also, there is a gap between the "deformation of the ambient space" described in Figure 1 and what is actually done. Relatedly, since g$(v, v) = <v, v> + \lambda <v, s>^2$, geodesics essentially minimize $<v, s>$. However, this vanishes whenever $v$ is orthogonal to $s$. In the very high-dimensional spaces where diffusion models operate, this condition imposes only a one-dimensional constraint. The proposed method may work for low-dimensional examples like that in Figure 2, where the manifold degenerates in one dimension, but its generality in high dimensions is unclear. In general, the data manifold has many degenerate directions, which span a normal space.

(2) This paper introduces an extrapolation method, but it does not produce geodesics and is inconsistent with the interpolation method.

### Presentation

(3) From Figure 4, all methods do not yield smooth interpolations. Interpolations between more similar images might be easier to assess.

(4) In Table 2, NoiseDiffusion achieves the best LPIPS, yet the score of the proposed method is emphasized, which is MISLEADING.

(5) For the metrics where the proposed method wins (FID and KID), the runner-up is LERP, and LERP is well known to perform poorly. Therefore, these metrics may be inappropriate. The closeness of LERP and SLERP scores also suggests these metrics are not sensitive to their differences. In that light, NoiseDiffusion may actually be superior. Thus, the paper does not convincingly demonstrate effectiveness on high-dimensional data.

### Literature

(6) There are prior works that define Riemannian metrics using the score function, but they are neither compared nor cited.

Yu et al., Probability Density Geodesics in Image Diffusion Latent Space, CVPR, 2025.
Saito et al., Image Interpolation with Score-based Riemannian Metrics of Diffusion Models, ICLR Workshop, 2025.

Overall, the background theory is interesting, and the proposal is promising. However, experiments and validations are insufficient, leaving the practical usefulness unclear.

**Questions:**

See Weaknesses.

---

### Official Review · Reviewer_U9ka · 2025-10-29

**Soundness:** 2
**Presentation:** 3
**Contribution:** 2
**Rating:** 4
**Confidence:** 3

**Summary:**

This paper proposes a method of constructing a Riemannian metric on the data manifold by using the learned score function of existing diffusion models, where the metric stretches along the normal direction of the manifold and preserves the tangential direction. This method requires no further parametrization for defining a metric on the data manifold. The paper shows smoother interpolations on synthetic sphere, Rotated-MNIST, and Stable Diffusion latents (MorphBench) over baselines (LERP/SLERP/NoiseDiff)

**Strengths:**

1. The method doesn't require additional training to obtain a metric on the data manifold given a pre-trained diffusion model. The authors also proposes an extrapolation method.
2. The method shows smoother interpolation and better performance than LERP/SLERP/NoiseDiff on synthetic sphere, Rotated-MNIST, and Stable Diffusion latents (MorphBench)
3. The authors shows some successful extrapolation examples

**Weaknesses:**

1. The paper proposes a method for defining a Riemannian metric on the data manifold for interpolation, but does not compare with many prior works that proposes methods for similar applications for data manifold interpolation (e.g. those listed in appendix A's related work section, https://arxiv.org/abs/2410.12779, GRAE, etc), especially methods that also learns a metric on the data manifold for interpolation (https://arxiv.org/abs/2405.14780, https://arxiv.org/abs/2410.04543). While these methods typically require an additional neural network for learning the metric, a comparison of performance v.s. computational cost would be informative.
2. The author's proposes an extrapolation method but only provides a handful of examples on the toy sphere dataset and rotated MNIST, which does not provide sufficient validation for the effectiveness on this method for more complicated data. There is also no comparison with whether prior methods could also achieve similar extrapolation.
3. The authors claim that this method could be used for understanding the geometric structure learned by diffusion models, but there isn't any further exploration beyond just data interpolation. It'll be nice to have additional demonstrations to substantiate this claim.

**Questions:**

1. How does your method compare with recently proposed manifold interpolation methods (see weakness 1)?
2. Does your model produce high quality extrapolation on more complicated data than toy datasets and rotated MNIST? Do prior methods fail in these cases?
3. Do the geodesics learned by your model result in better downstream applications that uses them, e.g. generative modeling?

---

### Official Review · Reviewer_dQue · 2025-10-31

**Soundness:** 2
**Presentation:** 2
**Contribution:** 2
**Rating:** 2
**Confidence:** 4

**Summary:**

This paper investigates the geometric structure implicitly learned by diffusion models and proposes a Riemannian metric derived directly from the model’s score function. Specifically, the authors define a local metric $g(x) = I + \lambda s(x) s(x)^{T}$ that induces a local geometry at $x_{t}$. This paper argues that this metric captures the anisotropy of the learned distribution, thereby providing a principled way to measure distances and explore geodesics, interpolation, and extrapolation. This manuscript visualizes the induced manifolds and investigates their alignment with semantic directions.

**Strengths:**

- The paper introduces a simple and clear formulation of a score-based Riemannian metric, offering a geometric interpretation of diffusion models.
- The proposed framework opens a promising direction for understanding the latent structure of diffusion models.
- The writing is generally clear and easy to follow.

**Weaknesses:**

- The proposed metric $g(x) = I + \lambda s(x) s(x)^{T}$ penalizes only a single direction along the score vector $s(x)$. When the data manifold has a codimension greater than one, normal directions orthogonal to $s(x)$ remain unpenalized. This limitation raises questions about the completeness of the metric in capturing local manifold curvature.
- The Geodesic Computations Algorithm requires heuristic techniques, such as adding the same noise in Line 223. Furthermore, the denoising step in Line 241 lacks sufficient description. It would be helpful to clarify whether deterministic solvers (e.g., DDIM or PF-ODE) are used.
- In Line 264, the score function $s(x_{i})$ is described as guiding the trajectory toward the data manifold. However, the score function is normal to the manifold, not tangential. The authors should clarify how this interpretation is consistent with the geometric intuition behind the proposed metric.
- The manuscript omits comparisons with existing interpolation methods such as [1, 2].
- The image interpolation results are evaluated only on MorphBench. Additional experiments on more diverse datasets, such as Animal Faces-HQ or CelebA-HQ as in [3], would strengthen the empirical validation and demonstrate the method’s generality.

[1] Yu, Qingtao, et al. "Probability density geodesics in image diffusion latent space." CVPR 2025.
[2] Samuel, Dvir, et al. "Norm-guided latent space exploration for text-to-image generation." NeurIPS 2023.
[3] Saito, Shinnosuke, and Takashi Matsubara. "Be Tangential to Manifold: Discovering Riemannian Metric for Diffusion Models." Arxiv.

**Questions:**

- How sensitive is the proposed metric to the diffusion noise level $t$? Does the induced geometry change significantly across timesteps, and if so, how should one select or aggregate across $t$?
- Could this metric be leveraged for practical applications?

---

### Note · Authors · 2025-12-01

**Comment:**

We sincerely thank all reviewers for their thorough evaluations and constructive feedback. We appreciate the time and effort invested in reviewing our work.

We acknowledge the key concerns raised, particularly regarding the limitation of our metric to penalizing only along the score direction and the need for more extensive experimental validation. We will address these issues in future iterations of this work.
Regarding the metric formulation, we agree that when the co-dimension is greater than one, other normal directions need to be taken into account.

However, we emphasize that the score vector represents the strongest normal direction, and our experiments showed promising results for this simplified approach. We acknowledge the limitations in our theoretical justification of this claim. Our primary goal was to introduce an accessible geometric framework for understanding diffusion models without requiring explicit parameterization—unlike pullback metrics, which require such manifold parameterization and remain computationally prohibitive for complex data manifolds such as natural images.

However, given the current ratings and the substantial revisions needed, we recognize that the scope of work required cannot be adequately addressed within the rebuttal period and would be unlikely to achieve acceptance thresholds. Therefore, we have decided to withdraw this submission.

We are grateful for the valuable feedback, which will substantially improve future versions of this work. We plan to incorporate comparisons with pullback metrics and provide stronger theoretical justification before resubmission.

Thank you again for your careful consideration.

**Withdrawal Confirmation:**

I have read and agree with the venue's withdrawal policy on behalf of myself and my co-authors.